# Prey Size Decline as a Unifying Ecological Selecting Agent in Pleistocene Human Evolution

**Miki Ben-Dor** * and **Ran Barkai** *

Department of Archaeology, Tel Aviv University, P.O.B, Tel Aviv 39040, Israel
* Correspondence: bendor.michael@gmail.com (M.B.-D.); barkaran@tauex.tau.ac.il (R.B.)

**Abstract:** We hypothesize that megafauna extinctions throughout the Pleistocene, that led to a progressive decline in large prey availability, were a primary selecting agent in key evolutionary and cultural changes in human prehistory. The Pleistocene human past is characterized by a series of transformations that include the evolution of new physiological traits and the adoption, assimilation, and replacement of cultural and behavioral patterns. Some changes, such as brain expansion, use of fire, developments in stone-tool technologies, or the scale of resource intensification, were uncharacteristically progressive. We previously hypothesized that humans specialized in acquiring large prey because of their higher foraging efficiency, high biomass density, higher fat content, and the use of less complex tools for their acquisition. Here, we argue that the need to mitigate the additional energetic cost of acquiring progressively smaller prey may have been an ecological selecting agent in fundamental adaptive modes demonstrated in the Paleolithic archaeological record. We describe several potential associations between prey size decline and specific evolutionary and cultural changes that might have been driven by the need to adapt to increased energetic demands while hunting and processing smaller and smaller game.

**Keywords:** human evolution; megafauna extinction; fat; domestication; human brain expansion; Paleolithic




## 1. Introduction

The potential role of human overhunting in megafauna (>45 kg) extinctions during the Pleistocene is a subject of long debate. However, the effect of megafauna extinctions on humans has been seldom discussed.

The genus *Homo* underwent an extensive set of physiological, cultural, and behavioral changes during the Pleistocene (roughly 2.6 million to 11.7 thousand years ago). At the end of this period, humans had established themselves as a species of unprecedented ecological dominance. Most notable among these changes was the directional increase in brain volume in the lineages leading to *H. sapiens*, the habitual use of fire, periodical change of stone-tool technologies, big-game hunting, resource intensification, food production, and animal and plant domestication.

We hypothesize that large prey's declining availability was a prominent agent of selection (sensu MacColl [1]) in human evolution and cultural change. We argue that *H. erectus* evolved to become a carnivore, specializing in large prey beginning 2 million years ago. Later, as prey size declined, humans adapted to acquire and consume smaller and smaller prey while adapting to maintain a constrained bioenergetic budget.

We first review the decline in prey size throughout the Pleistocene. We then discuss two sub-hypotheses at the base of the master hypothesis—1. acquiring animal-sourced food was critical to human survival and 2. humans preferred and adapted to acquire and consume large prey. The sub-hypotheses were presented in detail in three papers, which we briefly review here [2–4]. Having established the prey size decline and its potential effect on humans, we speculate on evolutionary and cultural adaptations in human prehistory that could have been caused by prey decline as an agent of selection.

Full testing of such a wide-ranging hypothesis requires many years of work, gathering and analyzing quantitative data about prey size dynamics in specific periods and places and quantifications of tempospatially associated specific evolutionary and cultural changes. Here, we present the hypothesis in broad brushstrokes with the intention of it generating interest and further exploration.

## 2. Pleistocene Decline in Prey Size

In recent years, it has become clear that the Late Quaternary megafauna extinction (LQE) is not the first megafauna extinction event that humans faced or caused. A long-term decline in megaherbivore (>1000 kg) diversity in Africa, beginning ~4.6 million years ago (Mya), was identified by Faith, Rowan, and Du [5]. Size 5 (>1000 kg) herbivore richness declined throughout the period, while size 4 (350–1000 kg) and size 3 (80 = 350 kg) herbivores began to decline around 1 Mya (Figure 2 in [5]). Faith et al. attributed the initial change to a drying climate and only later declines to *H. sapiens'* hunting pressure. All size-decline trends continued throughout the Pleistocene. Smith et al. [6] also identified a reduction in African terrestrial mammals' mean body weight during the Pleistocene, abruptly reversing a continuous growth trend of 65 million years. Particularly relevant to our hypothesis is that by the beginning of the Late Pleistocene, 126 thousand years ago (Kya), the mean body mass of mammals in Africa had declined to 50% of the expected value for such a large continent. This means that a substantial decline in the diversity and number of large herbivores occurred during the Middle Pleistocene. Smith et al. [6] attributed this substantial decline in diversity and number of large herbivores to the presence of the carnivorous humans on the continent during the Pleistocene (but see [7]).

In East Africa, a significant faunal turnover that resulted in prey size decline was identified between 500 and 400 Kya in Lainyamok, Kenya [8] and before 320 Kya in Olorgesailie, Kenya, during the period leading to the transition to the Middle Stone Age (MSA), and the subsequent appearance of *H. Sapiens* [9]. A continuous decline in the weighted mean mass of mammals in archaeological sites is also evident in the Levant starting at 400 Kya, where it is not associated with climate change [10,11]. The decline in megafauna continued or resumed globally during the Late Quaternary [12] and the Holocene [13]. In summary, in Africa, the Levant, and Europe, there was a continuous decline in prey size from the late Early and Middle Pleistocene, and a Late Pleistocene decline followed the arrival of *H. sapiens* to new continents and islands [14,15]. It is difficult not to feel that the temporal and geographical spread of the decline in the largest prey and its unidirectionality at each time and place is a result not of a changing factor (climate) but rather a constant factor (humans' preference for large prey). The current risk of extinction is also skewed towards larger fauna [16]. Studies of hunting by recent indigenous populations who rely on subsistence hunting show that they extend their hunts to smaller prey only when large prey got depleted. This behavior often results in declines and local extinctions of large-bodied mammals [17,18].

The debate over the anthropogenic nature of extinctions remains active [7,19,20].

Zooarchaeologists often question whether archaeological faunal assemblages reflect prey selection or prey abundance. Hypothesizing that humans specialized in large prey (see Section 4), a decline in prey size in the assemblages cannot reflect changing prey selection, because if large prey animals were abundant, people would preferentially acquire them (see [21] for recent support).

In a 2014 book chapter Wilkinson writes, "The first task of the prehistorian must be to decide which trophic level the population he is studying occupied" [22] (p. 544). A solid estimate of the human trophic level throughout the period that we discuss here is essential in order to judge the strength of the effect on humans of prey size decline.

## 3. The Trophic Position of Humans

We recently published a multidisciplinary review of the evidence regarding the human trophic level evolution based on 25 lines of evidence. We adapted a palaeobiological

approach, including human metabolism and genetics, body morphology, dental pathology, and life history. We also reviewed archaeological, ethnographic, paleontological, and zoological literature to identify changing patterns in faunal abundance, flora, lithic industries, stable isotopes, and other geoarchaeological data, and human behavioral adaptations to carnivory or omnivory that reflected past human trophic levels [4].

The review finds support for the notion that humans were carnivores starting from *H. erectus*. An analogy with other social carnivores indicates that carnivorous humans would have been hypercarnivores, consuming over ~70% of their calories from animal sources. A trend of declining trophic level (an increase in the plant component of the diet) is evident at the end of the MSA in Africa and the Upper Paleolithic (UP) period, and especially towards the end of the UP in the rest of the old world.

Figure 1 lists the evidence by human species and period. The full description of each line of evidence and a full list of references can be found in the source paper [4].

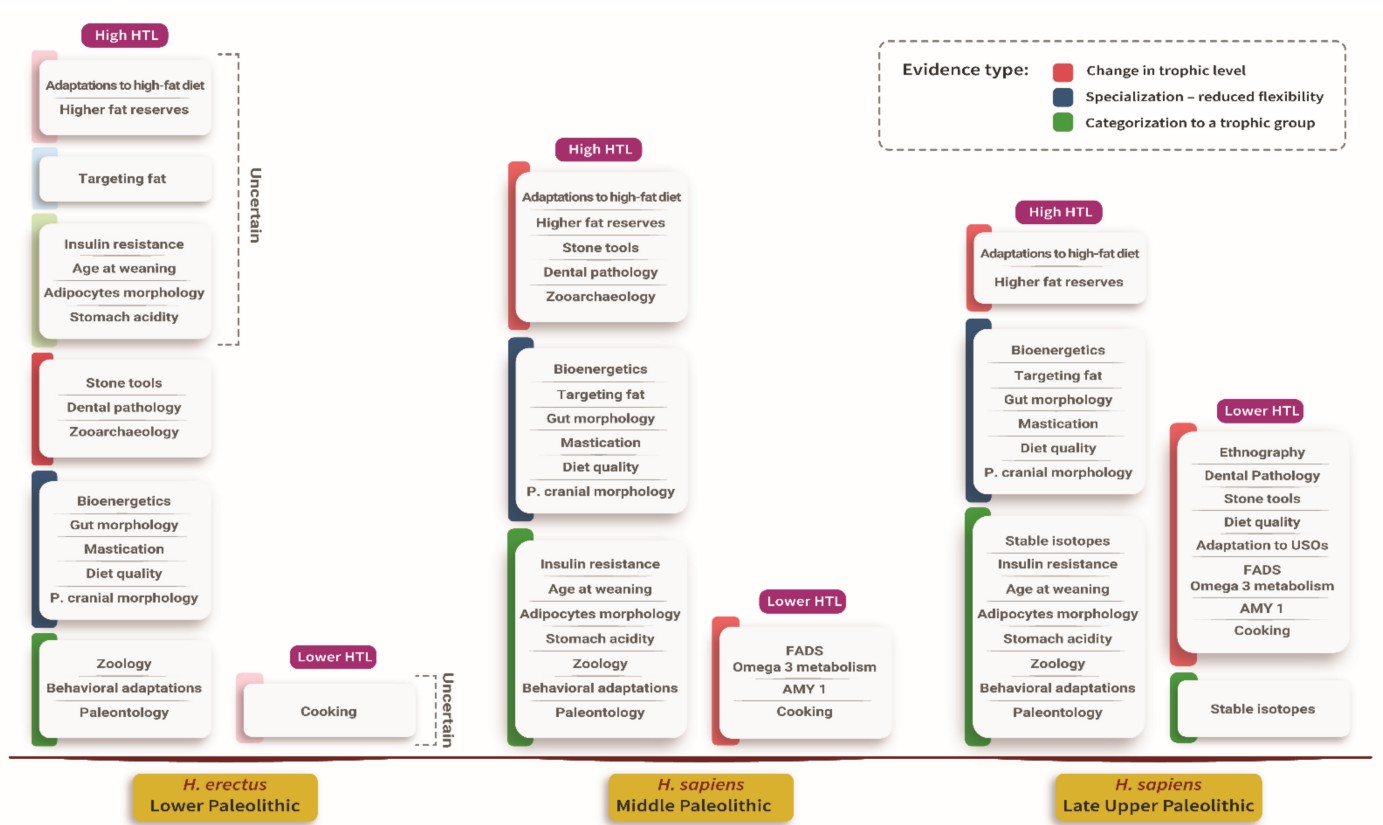

**Figure 1.** A list of evidence by trophic level, human species, period, and type of evidence. The evidence was divided into three types–1 (red): evidence for a change in the trophic level, 2 (blue): evidence for specialization, and 3 (green): categorization to a trophic group. Uncertain association of an item with *H. erectus* appears in muted color.

Figure 2 draws the trophic level route that humans experienced during the Pleistocene according to the thesis in [4].

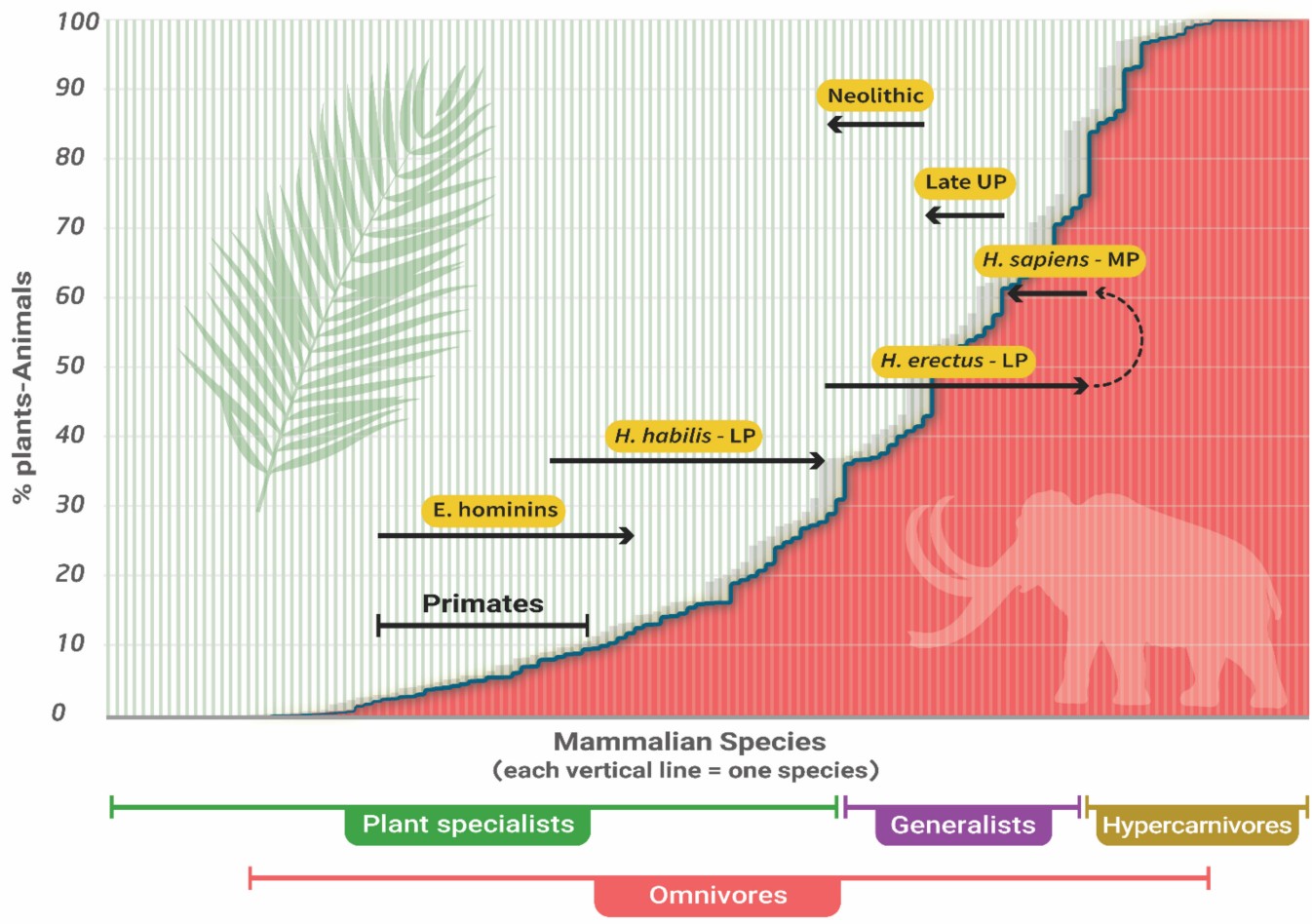

**Figure 2.** Proposed evolution of the human trophic level during the Pleistocene. LP–Lower Paleolithic; MP–Middle Paleolithic; UP–Upper Paleolithic; E. hominins–Early hominins, (Australopithecus, Paranthropus). Background and position of primates adapted from [23]. Each line corresponds to the plants and animals' food-source ratio of one mammalian species. Plant specialists and hypercarnivores–mammals that obtain over 70% of their food from plants and animals, respectively. Omnivores–any mammal that obtains food from both plants and animals.

### 4. Specialization in Large Prey

In [2], we claimed that 20th-century hunter-gatherers might be analogous in terms of ecological conditions and technology to humans at the end of the Paleolithic, after the Late Quaternary megafauna extinction, rather than to earlier Paleolithic humans. The late UP technology of bows and arrows, dogs, and grinding stones can be explained by the need to hunt smaller, fleeing animals and obtain an additional portion of the energy from plants at acceptable energetic costs. Thus, the 20th-century hunter-gatherers (HGs) hunting mix, dietary variability, and high plant consumption cannot be used as analogs for humans' diets in earlier Paleolithic periods.

In [4], we argued that large animals are underrepresented in archaeological assemblages. We analyzed an actual case containing 60 hunts of the Hadza to find that giraffes contributed 57% of the hunts' true total biomass, while their minimum number of individuals (MNI) was 14% of the total MNI and the number of identified specimens (NISP) was 7% of the total. However, only 8 of 11 giraffes were represented in the assemblage's MNI, therefore, weighting the MNI's by biomass still left a markable underrepresentation of the largest animal, pointing to potential substantial underrepresentation of very large animals such as elephants in archaeological assemblages.

Calculating and comparing the relative biomass abundance by prey size of a sample of sites from the Acheulian and Acheulo-Yabrudian, Early and Middle Stone Age, Middle Paleolithic Mousterian, Upper Paleolithic, and Aurignacian, we found that prey animals of a size above 200 kg provided 60–100% of the biomass in every sample. In each of the three comparisons, we found a decline in the relative biomass of the >200 kg animals in the later periods [4].

We reviewed four factors that made megaherbivores critically important to humans—1. High ecological biomass density, 2. Lower complexity of acquisition, 3. Higher net energetic return, and 4. High-fat content. Here we provide a summary of each factor; all references for the next section can be found in [2].

### 4.1. High Relative Biomass

Hempson et al. [24] estimated that 1000 years ago in Africa, the "nonruminants" group, which contains mainly megaherbivores, had a biomass density that is six times higher than the second densest group of "water-dependent grazers". They predicted that "elephants dominated the African herbivore biomass, often having biomasses equivalent to those of all other [herbivores] species combined". Even presently, after suffering extreme hunting pressure, studies find that megaherbivores, and in particular elephants, sustain high biomass densities. Elephants form 80–89% of the herbivores' biomass in several African nature reserves.

### 4.2. Not Escaping–Easier Tracking and Less Complex Hunting Tools

Megaherbivores do not rely on escape as a predator protection strategy, as evident by their low maximum speed compared to that of a lion (Figure 3). Unlike ungulates, megaherbivores lack specific predation risk alarm signals. When humans approach, they tend to stand still and may flee or charge when humans get closer. This behavior has several implications that make their acquisition by humans relatively energetically more profitable and technologically less complex (albeit not less courageous) than hunting smaller, fleeing prey. A common ethnographic method of hunting megaherbivores is to limit the movement in mud, a forest, or a pit, for example, and dispatch them with a thrusting spear [25,26]. Hunting faster, fleeing animals usually requires hunting from a distance by the use of more complex projectile weapons.

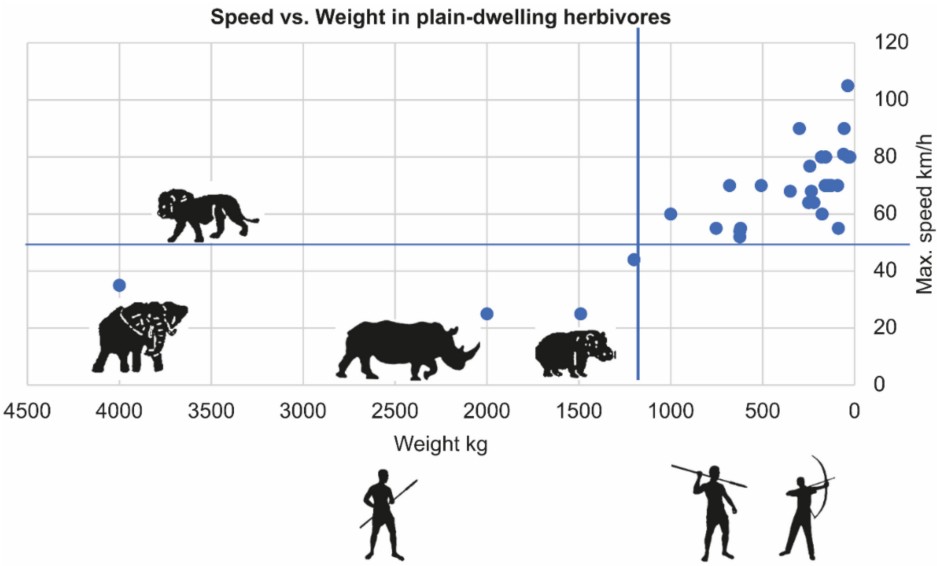

**Figure 3.** Hunting methods as a function of prey weight and maximum speed. Megaherbivores (>1000 kg) are slower than a lion (horizontal line), so they do not rely on escape as a predation prevention strategy. Data in [27]. Composite projectile weaponry is required to hunt smaller and faster animals.

*4.3. Larger Prey Contains Higher Body Fat Levels*

Protein consumption in humans is limited to around 35–50% of the daily calories due to the liver and kidney's limited ability to remove large quantities of the toxic nitrogen by-product of protein metabolism [28]. Thus, depending on the relative energetic returns and abundance of plants and animal fat, humans have to obtain 50–65% of their calories from either animal fat or plant fat and carbohydrates. According to ethnographic studies, the energetic return per hour on plant gathering is about one-tenth of medium-sized animals (Tables 3.3 and 3.4 in [29]); thus, it is expected that humans had an important obligatory requirement for animal fat [10].

Pitts and Bullard [30] were the first to find that larger mammals contain relatively more fat than smaller animals. Our analysis of a dataset of 257 animals from 19 African herbivore species [31] confirmed this phenomenon in Table 1 and Figure 4.

**Table 1.** Adjusted caloric fat percentage (E%) of African and non-African prey animals, based on [31] (Details of adjustments in [27]).

| Species | Weight Kg | Ledger–Raw Data (E%) | African Adjusted (E%) | Non-African Adjusted (E%) |
|---|---|---|---|---|
| Females | | | | |
| Hippo | 1277 | 67% | 71% | 76% |
| Wildebeest K | 192 | 62% | 67% | 71% |
| Waterbuck | 181 | 45% | 50% | 55% |
| Oryx | 161.5 | 57% | 62% | 67% |
| Wilderbeest S | 160.3 | 52% | 57% | 62% |
| Kongoni | 126.2 | 49% | 54% | 59% |
| Topi | 103.9 | 29% | 33% | 38% |
| Kobe | 62.1 | 45% | 49% | 55% |
| Warthog | 60.2 | 28% | 32% | 37% |
| Impala | 42 | 30% | 34% | 39% |
| Grant's Gazelle | 41.3 | 48% | 53% | 59% |
| Thomson's Gazelle | 18.4 | 34% | 38% | 43% |
| Thomson's Gazelle (S) | 16.9 | 47% | 52% | 57% |
| Males | | | | |
| Hippo | 1489 | 56% | 61% | 66% |
| Buffalo | 753 | 54% | 58% | 64% |
| Eland | 508.1 | 50% | 54% | 60% |
| Wildebeest K | 243.3 | 58% | 62% | 67% |
| Waterbuck | 237.7 | 20% | 23% | 27% |
| Wildebeest S | 203 | 64% | 68% | 72% |
| Oryx | 176.4 | 36% | 41% | 46% |
| Kongoni | 142.5 | 31% | 36% | 41% |
| Topi | 130.8 | 32% | 37% | 42% |
| Kobe | 96.7 | 34% | 39% | 44% |
| Lesser Kudu | 92.1 | 40% | 44% | 50% |
| Warthog | 87.8 | 26% | 30% | 35% |

**Table 1.** *Cont.*

| Species | Weight Kg | Ledger–Raw Data (E%) | African Adjusted (E%) | Non-African Adjusted (E%) |
|---|---|---|---|---|
| Grant's Gazelle | 60.1 | 36% | 40% | 45% |
| Impala | 56.7 | 29% | 33% | 38% |
| Gerenuk | 31.2 | 26% | 30% | 35% |
| Thomson's Gazelle | 25.3 | 31% | 35% | 40% |
| Thomson's Gazelle (S) | 20.3 | 30% | 34% | 39% |
| Average | 227 | 41% | 46% | 52% |

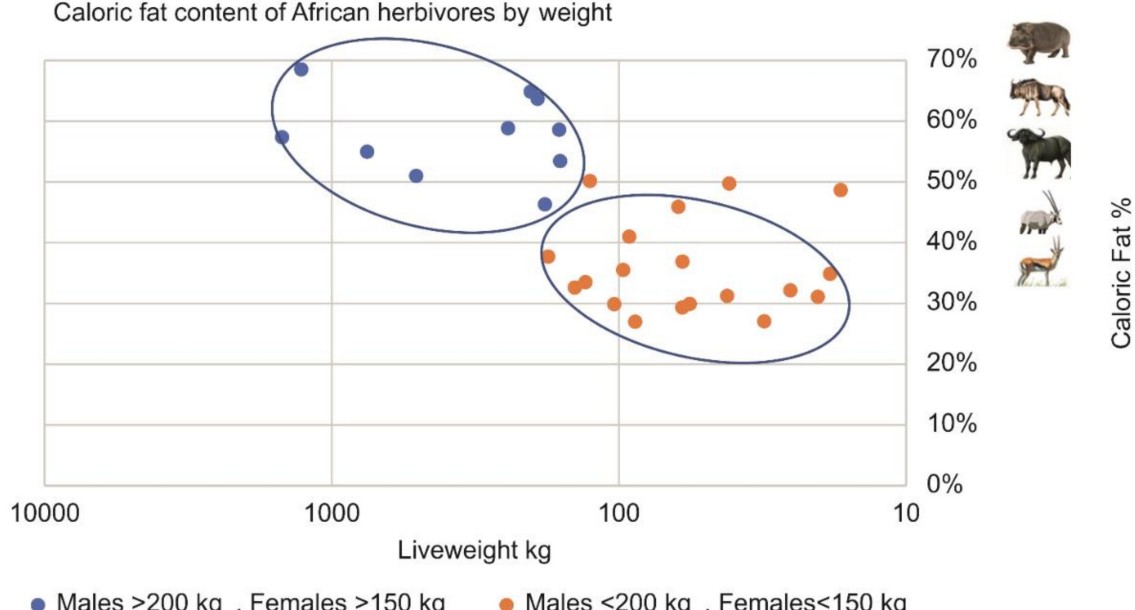

**Figure 4.** Percent of fat content as a function of body weight in African herbivores. Based on [31]. Data in [27].

In the dataset, male herbivores weighing over 200 kg and female herbivores weighing over 150 kg contained, on average, 44% more body fat relative to body weight than smaller animals. Large African herbivores such as hippo, buffalo, and eland contain 55–70% fat, while smaller herbivores such as impala and gazelle contain 30–35% fat. This means that the smaller animals' protein cannot be fully exploited unless plants or additional animals are acquired in which only the fat is consumed. In any event, to complete the obligatory fat requirement, a significant additional energetic expense is expected when the abundance of large animals declines.

Equally important, large herbivores lose less fat than smaller herbivores during periods of low forage. Thus, since humans mostly occupied seasonal environments, large animals became even more essential during low forage periods.

### 4.4. Larger Animals Provide a Higher Energetic Return

According to classic optimal foraging theory, an animal would specialize in the highest-ranking (highest energetic return) type if the encounter rate is high enough [32]. Data in [29] (Tables 3.3 and 3.4) show that medium-sized animals provide a net caloric return of 25–50,000 calories/hour compared to one-fifth to one-half of that in small animals (Figure 5); plant food returns are similar to those of very small animals. With such a difference in foraging efficiency, a decline in prey size also causes significant energetic pressure on humans.

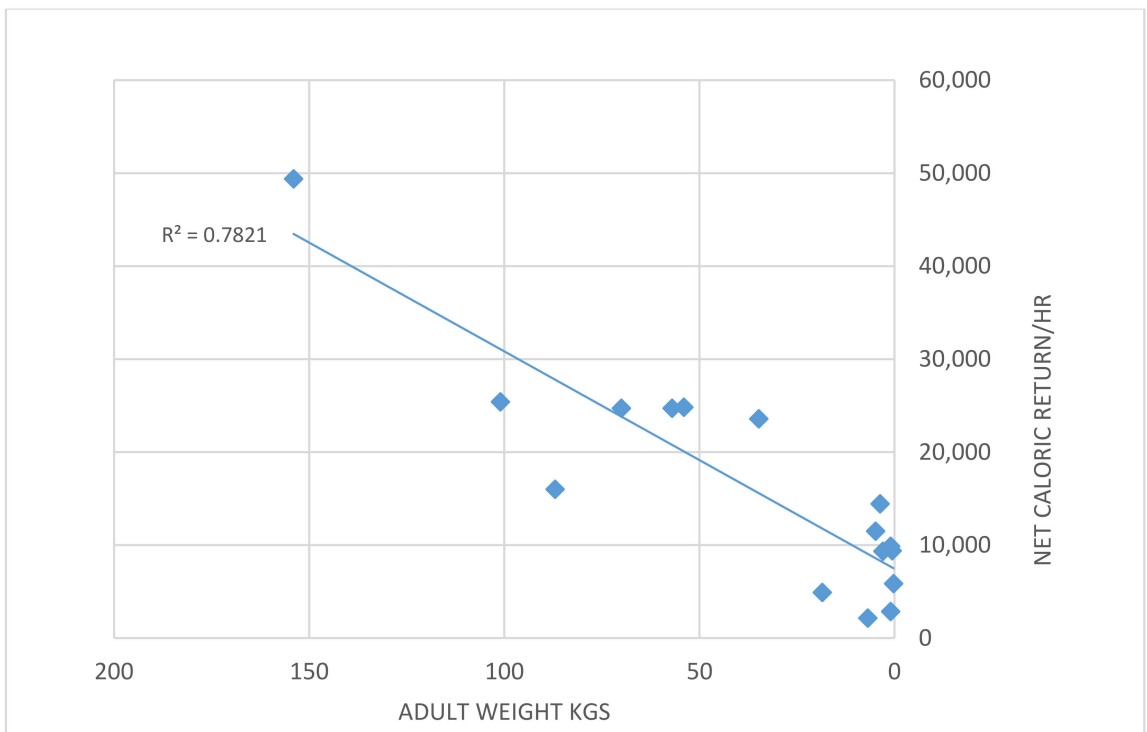

**Figure 5.** Net caloric return per hour as a function of prey size. Data in [27].

Opposition to the economic basis for large prey selection comes from the proponents of costly signaling as a choice criterion [33]. For example, based on data from the Hadza of Tanzania, Hawkes et al. [34] argued that the low hunting success rate and the need to share large prey prove that males hunt large game as a costly signal to attract mates. Wood and Marlowe [35], however, concluded, based on later data from the Hadza, that food economics rather than "show-off" signaling was at the base of the Hadza men's hunting preferences. Other researchers of recent hunter-gatherer groups also found that economic considerations are paramount in large prey selection [36]. In an archaeological context, Dominguez-Rodrigo et al. [37] reached a similar conclusion regarding the supremacy of economic considerations in the selection of large prey, based on analysis of the Upper Bed II, Olduvai Gorge (BK4B) faunal assemblage.

*4.5. Evidence for Specialization in Large Prey*

In [4], we explored evidence for humans' specialization in large prey acquisition, which is summarized in Table 2. All the references and detailed descriptions of the evidence can be found in the source paper.

**Table 2.** Evidence for humans' specialization in the acquisition of large prey during the Paleolithic.

| Evidence Name | Evidence Description |
|---|---|
| Bioenergetics | Large prey provides higher energetic returns per hour than smaller prey. The need to replace large prey with smaller prey is energetically costly. |
| Higher fat reserves | Humans have relatively high-fat reserves. Large prey is less abundant than smaller prey. Fat reserves may have evolved to allow extended fasting of several weeks, thereby bridging a variable encountering rate with large prey. Humans have adapted to efficiently synthesize ketones to replace glucose as an energy source for the brain during fasting. |

**Table 2.** *Cont.*

| Evidence Name | Evidence Description |
| --- | --- |
| Stomach acidity | Stomach acidity evolved, among other things, to guard against pathogens. Similar acidity level to scavengers in humans, higher than in carnivores, can be interpreted as an adaptation to a large prey's protracted consumption over days and weeks, whereby humans are acting as scavengers of their prey. |
| Targeting fat | Humans targeted fat by hunting large and prime-adult animals, both of which have a higher fat level, by bringing fatty parts to central places and exploiting bone fats at a great energetic expense. The recognition of targeting fat as a driver of human behavior supports the importance of large, higher fat bearing animals to humans' survival. |
| Stable isotopes | Researchers interpreted higher levels of nitrogen isotope 15 in humans than in carnivores as testifying to the higher consumption of large prey than other carnivores. |
| Paleontology | A decline in the guild of large prey carnivores 1.5 Mya was interpreted as resulting from humans' entrance to the guild. Moreover, the extinction of large prey throughout the Pleistocene is interpreted by some researchers as anthropogenic, testifying to humans' preference for large prey. |
| Zoological analogy | Large social carnivores get most of their energy from large prey. |
| Ethnography | Interpreting ethnographic and Upper Paleolithic technologies as an adaptation to smaller prey acquisition means humans were less adapted to smaller prey acquisition in earlier periods. |

## 5. Anthropogenic Contribution to Prey Size Decline

To the generally discussed arguments in support of anthropogenic prey-size decline, we would like to add the potential role of the need for fat to complete the non-protein portion of the diet in raising the risk for humans' prey extinction. In addition to the biased selection for large prey discussed in Section 4.3., the need for fat can also be satisfied by selecting prime-adults and partial consumption of only the fatty body parts of the prey.

Preference for hunting prime-adult animals has been identified as beginning 400 Kya [38] and perhaps 800 Kya [39], or even 1.8 Mya [40]. This phenomenon, which is unique among predators [41], is also prevalent in Neandertals' faunal assemblages throughout their wide-ranging habitat (e.g., [42–44]). Immature animals invest resources in growth at the expense of fat reserves [45]. Consequently, during most of the year, prime-adult animals will have a higher fat content than immature animals. Fat reserves also fluctuate differently between prime-adult males and females causing prime-adult females' preference during about half of the year (see [46] (Figure 5) regarding caribou).

Biased transport of fatty parts, including marrow-baring bones, is a common phenomenon in faunal assemblages indicating partial exploitation of prey [47–51].

Large prey is relatively more susceptible to extinction than smaller animals because of their low fecundity [52], and herds rely on a stable component of prime-adults, especially that of prime-adult females, for population stability [53]. Partial exploitation is a normal phenomenon in other carnivores in time of plenty [54] but the need for fat for humans is greater in time of ecological stress because of probable concomitant plant resources' scarcity, so it represents an increased risk for overhunting at times of stress for the prey populations [55]. To summarize this point, uniquely to humans, the need for fat increases

the predation pressure on vulnerable prey populations and thus the risk for extirpation or extinction of their prey.

It should be noted that the actual extirpation or extinction of specific prey species may depend on a local co-occurrence of high human predation pressure and stochastic external ecological deleterious conditions. Thus, spatial and temporal differences in the expirations and extinctions of the various prey species are expected, and thus the timing of the appearance of the humans' adaptations to the smaller prey species community.

## 6. The Decline in Prey Size as an Agent of Selection: Preliminary Case Studies

Given the significant difference in energetic return per hour between smaller and larger prey acquisition (Figure 5), human survival must have depended on adaptations that would mitigate the additional energetic cost of replacing the acquisition of extinct larger animals with smaller ones. Thus, we view the progressive decline in prey size as a selecting agent, and we view the progressivity of the adaptations as associated with that decline. We discuss several human biological, cultural, and behavioral transformations, and demonstrate how these might have been either dependent on large game availability or oriented toward mitigating the additional energetic cost of acquiring smaller prey during the Pleistocene.

### 6.1. Brain Size, Language, Stone-Tools, and Fire

The predominantly directional increase in brain size in the lineages leading to *H. sapiens* over more than two million years, during most of the Pleistocene (~2.6 Mya to ~0.3 Mya) and across several human species [56], is puzzling from an evolutionary theory point of view. A reversal of the growth trend at the end of the Pleistocene [57,58] also requires explanations. In present-day humans, larger cortical size is robustly associated with higher IQ [59]; a large brain relative to body mass has been shown to predict problem-solving ability in mammalian carnivores [60].

Increased social complexity was hypothesized to be the cognitive challenge that drove brain size growth [61]. Recently, however, ecological challenges, and in particular those related to foraging, have been proposed to better explain the need for brain expansion among primates [62–67]. A reduction in gut size, muscle mass, or redirection of energy from locomotion, growth, and reproduction may compensate for the increased energetic cost of a larger brain [65,68,69]. However, these compensations do not explain why a larger brain provided better fitness in the first place. Stanford and Bunn [70] proposed that the initial increase in the *Homo* brain size was driven by the need to develop hunting skills. Brain [71] attributed the brain size increase to the need to avoid predation; however, the question remains what drove the further ~50% increase in brain size from *H. erectus* to *H. sapiens*. Establishing the energetic pressure that the decline in prey size inflicted on humans, we propose that the expansion of various cognitive abilities met the ecological challenge of obtaining calories and fat from smaller prey at acceptable energetic costs. Brain expansion allowed humans to partly or wholly mitigate the potential additional energetic expenses on locomotion by tracking and linguistic communication of prey location, and facilitating economic smaller prey acquisition and exploitation by accumulating and transferring knowledge, and maintaining fire, and producing shaped and complex tools.

As indicated in several HG studies, movement on the landscape represents the largest discrete energetic expenditure of HG groups [72,73]. Therefore, tracking prey instead of relying on random encounters is a standard energy-saving behavior that could only have come about with an increase in cognitive skills, or the ability to deal with new information [74,75]. Blurton Jones and Konner [76] claimed that tracking is a cognitive process that mimics the scientific process, and used ethnographic research to argue that while tracking, hypotheses are formed and revised based on spoors' information.

Liebenberg [77] describes two methods of tracking—systematic and speculative. Systematic trackers track successive spoors, a conceivably more efficient strategy for tracking larger animals because they naturally leave more conspicuous signs of their passage and do

not flee (Figure 3). On the other hand, speculative trackers skip some potential spoors and proceed to where they speculate that the animal has headed, such as a water hole, an area of shade, or a food patch. Speculative tracking is more suitable for hunting smaller animals, which leave less conspicuous signs of their passage. Speculative tracking advances the hunter more rapidly on a shorter route and improves the tracking process's energetic efficiency. Liebenberg [78] states that "Speculative tracking requires much experience. So, most trackers start as systematic trackers and only become speculative trackers once they have mastered the basic skills". Additionally, the ability to identify fat-bearing animals, a critical ability when hunting smaller animals, also requires considerable experience and cognitive capacity [79] (pp. 42, 43).

Language is a large consumer of cognitive resources [80] and, hence, energy. We suggest that language increased fitness by facilitating energy savings in the face of prey size decline. Corballis [81] argues that language evolved in humans to communicate events "displaced in space and time from the present". A significant amount of energy can be saved by the quick and accurate exchange of information by group members about prey's recent sightings; information that could not be communicated appropriately without language.

Interestingly, bees use "dance language" to point to a food source that is not evenly distributed and displaced in space from where they are at the time [82]. In humans, the ability to also describe sighting time is essential as prey is more dynamic in the landscape than flower nectar. Additionally, language could help in the long-term retention and transfer of critical information concerning prey animals' behavior and countless details regarding the nature of the world in which hunters operate, all of which help save energy during tracking and hunting. Much of the fireside conversation of hunters' centers around natural phenomena and specific hunting experiences [76]. In summary, we propose that the evolution of a larger, energetically costly brain was driven to a significant extent by selection for energetic savings capabilities that secured smaller animals' acquisition at acceptable energetic costs.

Several researchers have claimed that increased mental capabilities facilitated technological innovations, such as the Lower Paleolithic cleavers or later multi-component projectile tools during the Pleistocene (e.g., [83–85]), and were most probably oriented toward the acquisition and processing of large game. The bow and arrow, atlatl, and fluted points [86] may represent inventions that were already improved by the initial expansion in *H. sapiens* brain size. These hunting technologies were mostly employed to target relatively small animals [26,87,88]. Transformations in stone-tool technologies could also be related to cognitive developments triggered by the need to acquire smaller and smaller prey.

The control of fire has been hypothesized as the reason for brain expansion in *H. erectus* [89]; however, evidence for fire's habitual use is much more common post-400 Kya (e.g., [90–92]. A central hearth that was continuously and intensively used is a prominent feature in the late Lower Paleolithic site of Qesem Cave, Israel (dated 420–200 Kya), where dental remains of post-*H. erectus* human lineage were discovered. Qesem Cave's faunal assemblage is dominated by the ~100 kg fallow deer (*Dama* cf. *mesopotamica*) and is devoid of elephants, common in earlier Lower Paleolithic sites [93]. It was argued that fire for roasting and cooking was intended to utilize the smaller animals more efficiently and was critical to the inhabitants' adaptation. The control of fire is considered part of a suite of innovative behaviors at Qesem Cave that demonstrate a new level of cognitive complexity, triggered by the disappearance of megaherbivores. One of these behaviors, the production of tiny sharp flint items utilizing lithic recycling to execute high-precision cutting tasks, was recently also associated with a new strategy for processing small game [94]. Moreover, the use of fire for roasting meat and extracting as many calories as possible from every food item continued progressively throughout the Paleolithic [95], correlating with the decline in prey size. Finally, sharing of smaller animals might have required a higher level of inhibitory control, another improved capability of a larger brain [67].

Neandertals also had a large brain, although they hunted large game alongside smaller animals. The comparison of cognitive abilities between Neandertals and *H. sapiens* is a sub-

ject of continuous research. There is little argument that Neandertals' brain structure was different, to some extent, from *H. sapiens* (e.g., [96–98], suggesting different functionality, which is the expected result of evolution under different ecological conditions.

Our mechanistic explanation for the correlation between the pace of brain growth and a decline in prey size during the Pleistocene can benefit from further testing. Initial indications of such a correlation can be found in East Africa where "brain expansion, independent of body size, appears to be most strongly expressed later, between 800 and 200 thousand years ago" [99] (p. 10), roughly correlating with a decline in prey size during the East African Middle Pleistocene [6,8,9,100].

Associating brain size increase with the mitigation of extra energetic costs that come with the need to hunt smaller prey can also explain the decline in brain size at the end of the Paleolithic period and beyond [57,58]. In that period, plant consumption increased [2], culminating in the domestication of plants and animals. Plants and domesticated animals do not escape so their acquisition does not require the same degree of knowledge and decision making under time pressure as hunting small prey does hence the lower cognitive requirements.

### 6.2. Hunting of Large Animals by H. erectus (sensu lato)

We concentrate here on the *H. erectus* (*sensu lato*) of Africa only and treat the species as a general representative of pre-*H. sapiens* species; as in most cases, due to the lack of human fossils, it is impossible to assign specific faunal assemblages to distinct pre-*H. sapiens* species. Determining that *H. erectus* had a carnivorous trophic level (Section 3) and accounting for the protein constraint (Section 4.3), it follows that *H. erectus* was dependent on large animals to provide the obligatory fat requirements. Although not related to the decline in prey size, this test case can help us understand the persistence of a mode of adaptation based on large animals' availability.

Recent analyses of the archeozoological and paleontological East African records portray *H. erectus* as a habitual hunter of large prey [101,102]. Preference for large prey animals during the Early Pleistocene is a conventional interpretation of archaeological assemblages [37,103–105]. Interestingly, Bunn and Gurtov [106] attribute to *H. erectus* a preference for prime-adult animal acquisition at FLK Zinji, 1.8 Mya. A similar preference is attributed by Bunn [107] to *H. heidelbergensis* in Elansfontain. Prime-adult animals always contain more fat than juveniles and older adults [45]; thus, it can be interpreted that this costly prey selection pattern was driven by a need for animal fat.

### 6.3. The Evolution of H. sapiens

The emergence of *H. sapiens* in Africa around 300 Kya [108] are contemporaneous with the onset of the Middle Stone Age mode of adaptation and, in East Africa, with the extinction of large-bodied grazing lineages and their replacement with related taxa of smaller body size [9]. Potts et al. [9] focused on the wet-dry climate variability, and the consequent need to cope with fluctuating resources as the drivers of changes at the onset of the MSA. However, we see carnivory specialization as a plausible solution to this and previous events of severe climate fluctuation [109]. Environmental variation can initiate specialization rather than flexibility in animals' behavior to reduce the experienced variation [110], as a predator's food sources are available in dry and wet conditions. Thus, the evolution of cognitive and cultural means of specializing in prey acquisition may be a viable and less costly solution to environmental variability than flexibility, which also has its costs [111]. In support, reviewing 1087 extant taxa from 28 phyla, Román-Palacios, Scholl, and Wiens [112] found that 63% are carnivores, and only 3% are omnivores. They state that their results "suggest that animals often specialize for carnivorous or herbivorous diet rather than being omnivores".

Regarding mammals, analysis of a large (N = 139) dataset of mammals' trophic levels [23] shows that 80% of the mammals in the dataset are omnivores, but most of the omnivores (75%) consume more than 70% of their food from either plants or animals,

leaving only 20% of the mammals in the dataset to be omnivore-generalists. Interestingly, while all of the 16 primates in the dataset were omnivores, 15 of the 16 were specialists. According to this somewhat counterintuitive point of view, the decline in prey size identified by Potts et al. [9] might be the most significant phenomenon in the transition to the MSA. As mentioned, an identical phenomenon, the appearance of a new human lineage and a new cultural complex temporally coupled with the disappearance of the largest herbivore (the straight-tusked elephant), is evident in the Levant 400 Kya [10]. The emergence of *H. Sapiens* in Africa, a new, post-*H. erectus* lineage in the Levant, and the concomitant new cultures in both places may represent adaptations to the acquisition and processing of smaller animals. Many physiological and behavioral characteristics of *H. Sapiens* may also have been directed toward saving energy when hunting prey.

The increase in brain size as an adaptation towards efficient tracking and hunting of smaller game has already been discussed. Increased locomotive energetic efficiency may have been achieved by the lighter, agile body, which produced long lower limbs relative to bodyweight [113,114]. Increased mobility can be a response to environmental variability as purportedly experienced at the onset of the MSA. For many animals, increased mobility "can functionally decrease environmental variation, especially if movement is coupled with choice behavior" [110] (p. 149). Thus, in humans, increased locomotive efficiency may have partly mitigated the additional energetic expenditures associated with hunting a greater number of smaller prey animals. Better locomotive efficiency also leads to an improved endurance running capability when hunting smaller, fleeing animals. However, it is possible that despite the more efficient locomotion, *H. sapiens* still had to adapt to higher metabolic expenses when prey size declined. The substantially greater basal metabolic rate and total energetic expenditures of *H. sapiens* [115] may be, in part, an adaptation to the additional energetic expenses that were imposed on *H. sapiens* by the need to obtain energy and fat from smaller prey.

Some of the face gracilization features in *H. sapiens* [116] may have also been enabled by the decline in prey size. Neandertals' robust frame has been attributed to the need to hunt large animals in close encounters [117], and it can be argued that a robust brow ridge is a part of this robusticity suite in pre-*sapiens*. Indeed, the reduced size of brow ridges in the *Homo* genus over time [116] could have been enabled by the decreased need to take down large animals at close encounters [26].

The habitual control of fire was discussed in the post-*H. erectus* context and applies to *H. sapiens* as well. It was also mentioned that the development of projectile technology by *H. sapiens* might have been intended for more energy-efficient hunting of smaller animals [26].

*6.4. The Extinction of the Neandertal*

Until recently, many researchers agreed that in Europe, the Neandertal diet had a narrow breadth and focused on larger prey [117–126]. A higher dietary plant content was postulated in more southern regions of the Neandertal's presence, such as the Levant [127,128]. Further, MIS 3 (~59–24 Kya) was a cold period leading to the Glacial Maximum, and cold regions such as tundra and taiga experience long periods of minimal vegetation, so it is reasonable to assume that Neandertals were also exposed to long periods of minimal vegetation in MIS 3 Europe.

Several researchers published a reconstruction of the Neandertal diet [121,124,129,130]. Large and medium-sized herbivores dominate the Neandertal archaeological faunal record in Europe, including proboscideans and rhinoceroses [51,120,121,131].

Stable isotope research (e.g., [118,123,126,132–135] unilaterally supports a carnivorous profile for the Neandertal diet in western Europe (but see discussion and some reservations in [129]). However, small animals and birds were also consumed by Neandertals (e.g., [136]).

In recent years, evidence for consumption of plants and cooking has emerged, based on plant residues in Neandertal dental plaque taken from fossils in Europe and Asia [124,137–139].

A single study of five sediment samples of Neandertal coprolites from El Salt (Spain), around 50 Kya, found that Neandertals predominantly consumed meat but also had a significant plant intake [140].

The Neandertal became extinct during Marine Isotope Stage 3 [141] in parallel with Europe's LQE [12]. Because of the Neandertal's heavier bodyweight and the cold weather, Neandertal total energetic expenditure (TEE) was estimated to be significantly higher than that of *H. sapiens* (e.g., [142]). In our model, higher TEE leads to higher obligatory fat consumption, especially in very cold, snow-covered conditions, when the availability of plant food is limited. For this reason, the Neandertal was more dependent on large animals with a high-fat level [143] that lose less fat compared to smaller animals during periods of low primary production [144]. Thus, in agreement with Geist [145] and Stewart [146], we hypothesize that the decline in prey size in Europe during the LQE was a significant driver of Neandertal extinction. It should be noted that there are many other hypotheses that attempt to explain the Neandertals' extinction. They cover cultural and other aspects of their complex way of living and some of them remain plausible and can co-exist side by side with ours.

### 6.5. Increased Plant Food Consumption from the Upper Paleolithic Onward

Although we know that plants were consumed, to some extent, whenever available (e.g., [147]), substantial archaeological evidence for increased plant consumption first appears in the Upper Paleolithic period [148–152]. The decreased availability of fat from large prey during the LQE may have led humans to develop technologies and behaviors that enabled them to obtain carbohydrates from plants as an alternative to animal fat, complementing their physiologically-limited protein consumption. Thus, we argue that the ubiquitous presence of plant food in post-Upper Paleolithic archaeological contexts is an effect of better preservation conditions and a reflection of the need to mitigate the energetic pressure and the reduction in fat availability due to the constant reduction in animal size.

### 6.6. Dog Domestication

Dogs were domesticated toward the end of the Pleistocene, during or after the LQE [153]. Since carnivores can utilize higher quantities of protein than humans [154], we agree with the hypothesis by Lahtinen et al. [155] that dog domestication was a form of "joint venture" between humans and wolves/dogs, in which humans contributed surplus meat protein from relatively fat-depleted animals that dogs could utilize but humans could not. In return, dogs helped humans save energy by helping to track and chase smaller animals. We add that in most ethnographic cases, dogs are employed to aid in hunting smaller animals [156,157]; thus, it is conceivable that dogs were domesticated as a behavioral adaptation to the increased energetic demands of hunting a larger number of smaller preys as prey size declined.

### 6.7. Plant and Animal Domestication at Different Times and Places

The continuing decline in the supply of animal fat during the Terminal Pleistocene may have led humans to domesticate plants and animals to secure an adequate supply of carbohydrates and fat to compensate for the ceiling on protein utilization. In the Levant, the small 25 kg *Gazella gazella* dominates the faunal assemblages of the Terminal Pleistocene, just before the Neolithic period and the Holocene, and large animals became negligible in comparison to previous periods [158,159]. A new pattern of hunting juvenile gazelles, which have a very low level of fat [45], then appeared [160]. Fat was extracted from gazelle bones [161], a possible nutritional stress marker [162] and, naturally, the cause for fat shortage. Local and global prey declines in the Late Paleolithic may explain domestication's appearance at this time. It may also explain this phenomenon's temporal variability and its independent appearance in different locations [163–165]. As previously stated, domestication and agriculture mark a clear departure from Paleolithic lifeways and must have necessitated a significant modification in work time, division of labor, and social

structure and hierarchy. Thus, resource intensification following these transformations must have been triggered by a forceful mechanism. Moreover, because humans interacted with animals and plants for many millennia before the advent of agriculture, ancient ecological knowledge was most likely available long before it was applied as a direct result of the continuous decline in prey size.

## 7. Conclusions

Our unifying hypothesis suggests one driver for many key physiological and cultural phenomena in human prehistory—the decline in prey size. Any unifying hypothesis is broad in scope and takes many years of testing before presenting a full set of supporting evidence and is also bound to touch on many long-standing debates in paleoanthropology. Here, we provided preliminary support for our contention that humans were hypercarnivores during most of the Pleistocene, starting with *H. erectus* and ending just before the end of the Pleistocene, possibly in the Neolithic. While the decline in prey size itself is well-documented, its temporal and geographical association with each explained phenomenon was proposed here with thick brushstrokes and should be extensively tested.

It is important to note that our proposed unifying ecological agent of selection was likely accompanied by other local ecological agents of selection for each studied evolutionary and cultural phenomenon that require identification. While we posit that the reduction in prey size indeed triggered human adaptation, we also wish to emphasize that some of the changes were fostered by a profound acquaintance with the environment coupled with ancient ecological and technological knowledge. As humans interacted with animals, plants, fire, and stone for over 3 million years, they became aware of the potential of these elements and could use this knowledge to survive in adverse circumstances. Moreover, we wish to clarify that the view presented in this paper is not deterministic, because humans may have played a central role in the prey size reduction by hunting large and medium-size mammals for hundreds of thousands of years, possibly cutting the branch on which they were sitting. Thus, these changes were not forced upon early humans but may have been an unavoidable human action outcome.

**Author Contributions:** Conceptualization, M.B.-D. and R.B.; methodology, M.B.-D. and R.B.; data curation, M.B.-D.; writing—original draft preparation, M.B.-D.; writing—review and editing, R.B.; visualization, M.B.-D. All authors have read and agreed to the published version of the manuscript.

**Funding:** This research received no external funding.

**Institutional Review Board Statement:** Not applicable.

**Informed Consent Statement:** Not Applicable.

**Data Availability Statement:** Data is available at Mendeley Data http://dx.doi.org/10.17632/mmjfj488j3.2.

**Acknowledgments:** We would like to thank Maria Rita Polombo for the opportunity to participate in this special issue and to the editors and the reviewers for their wise and timely contribution.

**Conflicts of Interest:** The authors declare no conflict of interest.

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
