# Peer review of "Prey Size Decline as a Unifying Ecological Selecting Agent in Pleistocene Human Evolution"

_quaternary, doi:10.3390/quat4010007_

Round 1

Reviewer 1 Report

This paper hypothesizes that the extinction of megafauna throughout the Pleistocene served as a selective force for modern human evolution. It is an interring hypothesis but the paper cannot be accepted for publication in it current form. The hypotheses needs to be developed further. 

One of the main issues in the paper in the current form is that the majority of their data comes from a paper in review by the same authors. Since this latter paper was not yet accepted for publication (and suggesting it is in review, would require a longer time for possible revisions), this paper appears to build extensively on that paper. Thus, the publication of this paper in its current form may in a sense circumvent the publication of the other paper.  Publication of this paper should and its revision should be built on a paper in press. 

However, if the above critique is put aside, there are other issues with the paper in its current form: 

  1. The idea that humans increased their plant consumption over time has no archaeological basis. The inherent bias in the archaeological record towards bone and stone tools compared to plant materials is extensive and cannot be taken as an assumption. 
  2. The argument that humans were hunters is still under debate. While it is probably accepted that Homo erectus hunted, it cannot be assume that a large selective source was scavenging, primarily  on an oppurtinistic base and the contribution, mirroring modern hunter gatherer societies, of gathering smaller species to the total caloric intake. 
  3. An interesting and very important point was made by John Speth in his book "Paleoanthropology and archaeology of big-game hunting" he argued for preference rather than abundance as the main criteria for faunal selection but not preference in the sense of HBE and energetics, as assumed here, but also cultural. The assumptions made in this paper need to further be discussed. 
  4. There a mismatch between the dates and locations, and the human species discussed. For a review that focuses on the entire Pleistocene, it mixes biological processes with the evolution of certain morphologies in humans across different time periods. It would be better to focus on the specific evolution of modern humans over the past 300,000 years rather than try to be all encompassing? 
  5.  I think that ignoring Neanderthal adaptations and morphology specifically in comparison to modern humans in misleading and lead to wrong conclusions. Indeed, the difference in energetics between the two species may reveal critical insights. 
  6. While this paper assumes that the issue of whether humans caused the megafaunal extinction or did climate create the conditions for it is not relevant to their hypotheses, I find that it is and needs to be discussed. For example, if humans led (albeit subconsciously) to megafaunal extinction then it would be interesting to address this issue as part of NCT but if climate were the so called culprit, they perhaps we have an issue of parallel selective forces rather than a cause and effect? 
  7. There are many assumptions/facts made that are not supported nor discussed such as the linear increase in brain size (Neanderthal brains are larger than are own, modern brain size is smaller than Pleistocene humans) is one example. 

Author Response

Point 1: One of the main issues in the paper in the current form is that the majority of their data comes from a paper in review by the same authors. Since this latter paper was not yet accepted for publication (and suggesting it is in review, would require a longer time for possible revisions), this paper appears to build extensively on that paper. Thus, the publication of this paper in its current form may in a sense circumvent the publication of the other paper.  Publication of this paper should and its revision should be built on a paper in press. 

Response 1: We are glad to advise that the in January 29th the paper has been accepted for publication at the Yearbook of Physical Anthropology (the annual supplement of the American Journal of Physical Anthropology, IF 2.4). As it should be online in a matter of few weeks at the most, we cite it as Ben-Dor, Sirtoli & Barkai, 2021. The reviewer is right to say that the Yearbook paper's data is one of the cornerstones of this paper. We would be glad to send a copy of the paper for the reviewer reading as we feel that many of the objections that he or she raises can be alleviated after reading the paper. As the paper has been accepted and will be published soon, readers could refer to it in case of need. In any case, we strongly believe that the summary of the data from that paper as presented in our current submission is solid, comprehensive and sufficient as a background for out hypothesis.

Point 2: The idea that humans increased their plant consumption over time has no archaeological basis. The inherent bias in the archaeological record towards bone and stone tools compared to plant materials is extensive and cannot be taken as an assumption. 

Response 2: We argued that humans increased their plant consumption markedly at the end of the Pleistocene, not "over time". We were not the first to notice the change as can be learned in the 5 citations in support the argument (line 551). Moreover, we strongly believe that the appearance and growing number of griding stones and harvesting tools towards the terminal Pleistocene could serve as evidence for the growing importance of vegetal foods in human diet during the Upper and Epi Paleolithic periods.

Point 3. The argument that humans were hunters is still under debate. While it is probably accepted that Homo erectus hunted, it cannot be assume that a large selective source was scavenging, primarily on an opportunistic base and the contribution, mirroring modern hunter gatherer societies, of gathering smaller species to the total caloric intake. 

Response 3: Debates are part of science. We feel that we did as much as necessary, if not more, to support our position with the publication of our paper on the evolution of the trophic level and summarizing the paper here. We strongly believe that the evidence for hunting and/or primary access of humans to animal carcasses during the Lower Paleolithic is generally acceptable nowadays (see, for example: Thieme, H. (1997). Lower Palaeolithic hunting spears from Germany. Nature, 385(6619), 807-810.‏ ; Dominguez-Rodrigo, M., & Pickering, T. R. (2017). The meat of the matter: an evolutionary perspective on human carnivory. Azania: Archaeological Research in Africa, 52(1), 4-32.‏ ; Carvalho, S., Thompson, J., Marean, C., & Alemseged, Z. (2019). Origins of the human predatory pattern: The transition to large-animal exploitation by early hominins. Current Anthropology, 60(1).‏ ; Bunn, H. T. (2019). Large ungulate mortality profiles and ambush hunting by Acheulean-age hominins at Elandsfontein, Western Cape Province, South Africa. Journal of Archaeological Science, 107, 40-49.). It goes without saying that alongside hunting humans might have scavenged as well, but we see this as a complementary aspect to hunting and thus do not deal with that in this paper. Additionally, we cite another paper, Ben-Dor & Barkai 2020, in which we thoroughly reviewed the problematic nature of the ethnographic record in reconstructing the trophic level of humans during the Palaeolithic period.

Point 4: An interesting and very important point was made by John Speth in his book "Paleoanthropology and archaeology of big-game hunting" he argued for preference rather than abundance as the main criteria for faunal selection but not preference in the sense of HBE and energetics, as assumed here, but also cultural. The assumptions made in this paper need to further be discussed. 

Response 4: We have added a paragraph in section 4.4. (lines 245-255). We do not concur with Speth on this point and feel it is lightly supported. We presented his point of view as well as those opposing it.

Point 5: There a mismatch between the dates and locations, and the human species discussed. For a review that focuses on the entire Pleistocene, it mixes biological processes with the evolution of certain morphologies in humans across different time periods. It would be better to focus on the specific evolution of modern humans over the past 300,000 years rather than try to be all encompassing? 

Response 5: The decline in prey size began before the appearance of H. sapiens. We chose to explain the H. erectus consumption of large prey to provide a starting point for the process. Otherwise, we only listed phenomena that pertain to H. sapiens and Neandertals.

Point 6. I think that ignoring Neanderthal adaptations and morphology specifically in comparison to modern humans in misleading and lead to wrong conclusions. Indeed, the difference in energetics between the two species may reveal critical insights. 

Response 6: We discuss TEE's difference between Neanderthals and H. sapiens in the original submission (lines 535).

Point 7. While this paper assumes that the issue of whether humans caused the megafaunal extinction or did climate create the conditions for it is not relevant to their hypotheses, I find that it is and needs to be discussed. For example, if humans led (albeit subconsciously) to megafaunal extinction then it would be interesting to address this issue as part of NCT but if climate were the so called culprit, they perhaps we have an issue of parallel selective forces rather than a cause and effect? 

Response 7. We accept the remark. Please see lines 101-108 and section 5 (line 263), where we proposed that anthropogenic factors contributed to the increased risk of megafaunal extinction and eventually, in combinations with external ecological conditions, led to extinction or expirtations. We chose not to use NCT in our explanation.

Point 8. There are many assumptions/facts made that are not supported nor discussed such as the linear increase in brain size (Neanderthal brains are larger than are own, modern brain size is smaller than Pleistocene humans) is one example. 

Response 8. We didn't write that the brain size increase was linear but directional. We added a citation for this statement. The decline in brain size toward the end of the Pleistocene was discussed and supported in the original version (line 421).

Reviewer 2 Report

The potential role of humans' overhunting in large animal extinctions during the Pleistocene is a subject of long debate. However, the opposite side that is the effect of megafauna extinctions on humans has been rarely discussed.

The authors hypothesize that large prey's declining availability was a prominent agent in humans' evolution and cultural change. We argue that H. Erectus became a carnivore specializing in a large game beginning two million years ago. Later, as prey size declined, humans adapted to acquire and consume smaller and smaller prey while maintaining constrained bioenergetic budgets.

These ideas are very well presented in a theoretical contribution with fascinating ideas and suggestions about human evolution and the effects of faunal changes on modern humans' emergence.

This paper is an excellent contribution to discussing this bidirectional effect by measuring several factors in a very innovative discussion. The critical aspect refers to whether the change in the large faunal extinction is a consequence of hunting intensity or other factors. If both elements are correlated, cultural responses to the decline of megafauna should differ if non-human factors mainly produced the extinction. Human adaptability may introduce different evolutionary traits and responses depending on the effect of human behavior itself.

On the other hand, the possible existence of economic resilience models that may occur in the case of the very late Acheulian of Iberia could question the idea of this mass/energetic correlation of the human evolutionary model. Recent contributions indicate that close to MIS5 different areas of the Iberian peninsula still maintain a "classic Acheulian" exploitation model.

I appreciate the perspective of this contribution. However, I consider that the interpretation falls into a very unidirectional view. This aspect is clearly appreciated in the 5.4 section about the neanderthal extinction. The authors simplify this process to the incapacity of anatomical adaptation and the consequent decrease of megafauna. Even if this perspective is plausible, this human group's history is so complex, and its extinction may also include cultural reasons as its motor.  In the conclusion and discussion section, the authors confirm the concurrence of other local evolutionary and cultural phenomenon in the general process. Perhaps, the ideas presented in conclusion could also be reflected in previous sections.

I also recommend some minor changes, particularly concerning the figures. I propose clarifying Figure 3, hunting methods as a function of prey weight and maximum speed. Data in Ben-Dor (2020), since it is not clear the assignation of species to the mass and/or speed (see the lion figure located in the same level as an elephant).

Apart from those minor aspects, I strongly suggest the publication of this paper since I consider it very valuable and provide relevant information for researchers.

Author Response

Point 1: The critical aspect refers to whether the change in the large faunal extinction is a consequence of hunting intensity or other factors. If both elements are correlated, cultural responses to the decline of megafauna should differ if non-human factors mainly produced the extinction. Human adaptability may introduce different evolutionary traits and responses depending on the effect of human behavior itself.

On the other hand, the possible existence of economic resilience models that may occur in the case of the very late Acheulian of Iberia could question the idea of this mass/energetic correlation of the human evolutionary model. Recent contributions indicate that close to MIS5 different areas of the Iberian peninsula still maintain a "classic Acheulian" exploitation model.

Response 1: We added a special section (section 5, line 263) to comply with point 1. We accept that the effect of prey-size reduction is not identical in anthropogenic or external causation cases. We added support to the anthropogenic view and explained why there would be a difference between areas in the pace of change, as is witnessed in Iberia during MIS 5.

Point 2: I consider that the interpretation falls into a very unidirectional view. This aspect is clearly appreciated in the 5.4 section about the neanderthal extinction. The authors simplify this process to the incapacity of anatomical adaptation and the consequent decrease of megafauna. Even if this perspective is plausible, this human group's history is so complex, and its extinction may also include cultural reasons as its motor.  In the conclusion and discussion section, the authors confirm the concurrence of other local evolutionary and cultural phenomenon in the general process. Perhaps, the ideas presented in conclusion could also be reflected in previous sections.

Response 2: We concur with the remark and have added a sentence at the end of the section to express that agreement. (line 544)

Point 3. I also recommend some minor changes, particularly concerning the figures. I propose clarifying Figure 3, hunting methods as a function of prey weight and maximum speed. Data in Ben-Dor (2020), since it is not clear the assignation of species to the mass and/or speed (see the lion figure located in the same level as an elephant).

Response 3: It is difficult to change the figure at this stage; however, we add to the figure description a note that clears possible misunderstanding.

Reviewer 3 Report

The well-written manuscript “Prey size decline as a unifying ecological selecting agent in Pleistocene human evolution” propose the hypothesis that Pleistocene Homo species specialised in the hunting of large prey and that the decline throughout the Pleistocene of the mammals with large body size engendered evolutionary and cultural adaptations in the human lineage. This very interesting hypothesis is worthwhile to be published. It can form a starting point for new analyses and can fuel the debate on human evolution.

I only have some minor comments.

Page 6 Larger prey contains higher body fat levels

Could the authors add references on the limit of protein consumption in humans?

It would be interesting to add here a table that details the fat content of large and medium prey mammals.

Page 10

Typos:

H. Erectus -> H. erectus

Knowladge -> knowledge

Censu lato -> sensu lato

Page 11

Typo:

H. Sapiens -> H. sapiens

Page 12

Typos:

H. Sapiens -> H. sapiens

H. Erectus -> H. erectus

Page 13 Dog domestication

It would be interesting to add here the recent reference of Lahtinen et al. (2021) on excess protein meat and its meaning on dog domestication.

Lahtinen, M., Clinnick, D., Mannermaa, K. et al. Excess protein enabled dog domestication during severe Ice Age winters. Sci Rep 11, 7 (2021). https://doi.org/10.1038/s41598-020-78214-4

2.12.0.0

2.12.0.0

2.12.0.0

2.12.0.0

Author Response

Point 1: Page 6 Larger prey contains higher body fat levels

Could the authors add references on the limit of protein consumption in humans?

Response 1: Added Speth & Spielman 1983 as reference (line 209)

Point 2: It would be interesting to add here a table that details the fat content of large and medium prey mammals.

Response 2: We added the table as Table 1 (line 218)

Point 3: Page 13 Dog domestication

It would be interesting to add here the recent reference of Lahtinen et al. (2021) on excess protein meat and its meaning on dog domestication.

Lahtinen, M., Clinnick, D., Mannermaa, K. et al. Excess protein enabled dog domestication during severe Ice Age winters. Sci Rep 11, 7 (2021). https://doi.org/10.1038/s41598-020-78214-4

Response 3: We added reference to the Lahtinen et al. 2021 paper, which we were happy to see. (line 566)

Round 2

Reviewer 1 Report

I appreciate the response of the authors to the comments and feel that they addressed my concerns. While I believe there are further organizational and language editing that I improve the MS, I do not see any impediment to it being accepted for publication in its current form.